# Prognosis of Extended-Spectrum-Beta-Lactamase-Producing Agents in Emphysematous Pyelonephritis-Results from a Large, Multicenter Series

**DOI:** 10.3390/pathogens11121397

**Published:** 2022-11-23

**Authors:** José Iván Robles-Torres, Daniele Castellani, Hegel Trujillo-Santamaría, Jeremy Yuen-Chun Teoh, Yiloren Tanidir, José Gadú Campos-Salcedo, Edgar Iván Bravo-Castro, Marcelo Langer Wroclawski, Santosh Kumar, Juan Eduardo Sanchez-Nuñez, José Enrique Espinosa-Aznar, Deepak Ragoori, Saeed Bin Hamri, Ong Teng Aik, Cecil Paul Tarot-Chocooj, Anil Shrestha, Mohamed Amine Lakmichi, Mateus Cosentino-Bellote, Luis Gabriel Vázquez-Lavista, Boukary Kabre, Ho Yee Tiong, Lauro Salvador Gómez-Guerra, Umut Kutukoglu, Joao Arthur Brunhara Alves-Barbosa, Jorge Jaspersen, Christian Acevedo, Francisco Virgen-Gutiérrez, Sumit Agrawal, Hugo Octaviano Duarte-Santos, Chai Chu Ann, Wei Sien Yeoh, Vineet Gauhar

**Affiliations:** 1Department of Urology, Hospital Universitario “Dr. José Eleuterio Gonzalez”, Monterrey 64460, Mexico; 2Urology Unit, Azienda ospedaliero-Universitaria Ospedali Riuniti di Ancona, Università Politecnica delle Marche, 60126 Ancona, Italy; 3Department of Urology, Hospital Covadonga, Córdoba 94560, Mexico; 4S.H. Ho Urology Centre, Department of Surgery, Prince of Wales Hospital, The Chinese University of Hong Kong, Hong Kong 999077, China; 5Department of Urology, Marmara University School of Medicine, Istanbul 34854, Turkey; 6Department of Urology, Hospital Central Militar, Ciuada de Mexico 11200, Mexico; 7Department of Urology, Hospital Israelita Albert Einstein, BP—a Beneficência Portuguesa de São Paulo, Sao Paulo 05562-900, SP, Brazil; 8Faculdade de Medicina do ABC, Santo André 09060-870, SP, Brazil; 9Department Urology, Christian Medical College, Vellore 632004, India; 10Department Urology, Hospital General de México “Dr. Eduardo Liceaga”, Mexico City 06720, Mexico; 11Hospital Regional de Alta Especialidad de la Peninsula de Yucatán, Merida 97133, Mexico; 12Department Urology, Asian Institute of Nephrology and Urology, Hyderabad 500082, India; 13Department Urology, King Abdulaziz National Guard Hospital, Riyadh 14611, Saudi Arabia; 14Division of Urology, Department of Surgery, University of Malaya, Kuala Lumpur 59100, Malaysia; 15Department of Urology, Hospital Central Military, Mexico City 11200, Mexico; 16Department of Urology, National Academy of Medical Sciences, Bir Hospital and B&B Hospital, Gwarko, Lalitpur 44700, Nepal; 17Department of Urology, University Hospital Mohammed the VIth of Marrakesh, Marrakesh BP2360, Morocco; 18Department of Urology, Federal University of Paraná, School of Medicine, Department of Urology, Hospital das Clínicas, 80060 Curitiba, PR, Brazil; 19Department of Urology, Hospital Médica Sur, 14050 Mexico City, Mexico; 20Department of Urology, Hospital Yalgado Ouedraogo Ouagadouga, Kadiogo 9FMV, Burkina Faso; 21Department of Urology, University Surgical Cluster, National University Hospital, Singapore 119074, Singapore; 22Department of Urology, Hospital Das Clínicas da Universidade de São Paulo, Sao Paulo 05402-010, SP, Brazil; 23Department of Urology, AIIMS Bhubaneshwar, Sijua, Patrapada, Bhubaneswar, Odisha 751019, India; 24Department of Urology, Hospital Municipal Dr. Moysés Deutsch (M’Boi Mirim), Sao Paulo 04849-030, SP, Brazil; 25Department of Minimally Invasive Urology, Ng Teng Fong General Hospital, Singapore 609606, Singapore

**Keywords:** emphysematous pyelonephritis, extended-spectrum beta-lactamases, prognosis, nephrectomy, minimally invasive procedures

## Abstract

Background: Emphysematous pyelonephritis (EPN) is a necrotizing infection of the kidney and surrounding tissues with significant mortality. We aimed to assess the clinical factors and their influence on prognosis in patients being managed for EPN with and without ESBL-producing bacteria and to identify if those with EPN due to ESBL infections fared any different. Methods: A retrospective analysis was performed on patients with EPN diagnosis from 22 centers across 11 countries (between 2013 and 2020). Demographics, clinical presentation, biochemical parameters, radiological features, microbiological characteristics, and therapeutic management were assessed. Univariable and multivariable analyses were performed to determine the independent variables associated with ESBL pathogens. A comparison of ESBL and non-ESBL mortality was performed evaluating treatment modality. Results: A total of 570 patients were included. Median (IQR) age was 57 (47–65) years. Among urine cultures, the most common isolated pathogen was *Escherichia coli* (62.2%). ESBL-producing agents were present in 291/556 urine cultures (52.3%). In multivariable analysis, thrombocytopenia (OR 1.616 95% CI 1.081–2.413, *p* = 0.019), and Huang–Tseng type 4 (OR 1.948 95% CI 1.005–3.778, *p*= 0.048) were independent predictors of ESBL pathogens. Patients with Huang–Tseng Scale type 1 had 55% less chance of having ESBL-producing pathogens (OR 1.616 95% CI 1.081–2.413, *p* = 0.019). Early nephrectomy (OR 2.3, *p* = 0.029) and delayed nephrectomy (OR 2.4, *p* = 0.015) were associated with increased mortality in patients with ESBL infections. Conservative/minimally invasive management reported an inverse association with mortality (OR 0.314, *p* = 0.001). Conclusions: ESBL bacteria in EPN were not significantly associated with mortality in EPN. However, ESBL infections were associated with poor prognosis when patients underwent nephrectomy compared conservative/minimally invasive management.

## 1. Introduction

Emphysematous pyelonephritis (EPN) is a fulminant renal infection caused by gas-forming organisms that induce parenchymal destruction [1]. This is often seen in patients with compromised immune response such as those with diabetes mellitus (DM), chronic kidney disease (CKD), chronic steroid users, HIV, and renal transplant [2]. Extended-spectrum-β-lactamase (ESBL)-producing bacteria are Enterobacteriaceae, such as *Escherichia coli*, *Klebsiella* spp., and *Proteus* spp., that develop resistance against many antibiotics considered first-line treatment. Infections produced by ESBL-producing pathogens are associated with higher mortality than corresponding infections due to non-ESBL pathogens, with reported mortality rate between 3.7 and 22.1% [3,4,5].

The timely start of appropriate antibiotics and percutaneous catheter drainage (PCD) are important factors to improve prognosis in EPN patients [6,7]. Most studies usually incorporate β-lactamase inhibitors, cephalosporins, aminoglycosides, and quinolones as first-line treatment [8]. Increasing global antimicrobial resistance and inappropriate initial antibiotic therapy in ESBL infections could result in worse prognosis in this population. Outcomes of ESBL in EPN were first reported by Robles-Torres et al. [9]. They found that ESBL-producing EPN was not associated with a worse prognosis or association with increased mortality. However, this study relied on a single-center small cohort of patients and the hypothesis that ESBL infections could have a predilection for a specific set of patients and a potential association with worst outcomes remains unproven. The aim of this study was to determine the clinical factors and their influence on prognosis in patients being managed for EPN with and without ESBL-producing bacteria and to identify if those with EPN due to ESBL infections fared any different.

## 2. Materials and Methods

Prospectively collected databases from 22 centers across 11 countries were retrospectively reviewed for patients diagnosed with EPN between 2013 and 2020. Patients’ management was based on resources, experience, and protocols of the individual institutions. Inclusion criteria were age ≥ 18 years, signs and symptoms of upper urinary tract infection due to EPN and confirmed by CT scan. Patients with sepsis refractory to conservative or minimally invasive management (MIM) underwent nephrectomy. Conservative management was defined as supportive therapy, including fluid resuscitation, metabolic control, and broad-spectrum antibiotics. MIM included ureteral stent placement with or without percutaneous drainage of abscess or perinephric gas. Early nephrectomy was defined as surgery performed within 72 h of hospital admission. Patients with missing data and previous urinary tract instrumentation within three months of presentation were excluded. We gathered the following data upon admission: age, gender, comorbidities, clinical characteristics, and laboratory workup (complete blood count, blood chemistry, and urine culture) at presentation. We considered the following cutoff values: anemia as <12 g/dL hemoglobin, leukocytosis as >11,000/μL white blood cells, thrombocytopenia as <150,000/μL platelet, increased creatinine as a serum creatinine level ≥1.2 mg/dL, and hyperglycemia as serum glucose level >200 mg/dL. Data on urine cultures were acquired for analysis based on the guidelines of the Clinical and Laboratory Standards Institute. Identification of isolates were obtained by matrix-assisted laser desorption–ionization time-of-flight mass spectrometry (MALDI-TOF MS) [10]. The production of ESBL was performed with a double-disc sensitivity test. The qSOFA score was used to assess the risk of in-hospital mortality. The qSOFA score is a 3-point scoring system that includes altered mental status, >22 breaths/min, and systolic blood pressure <100 mmHg (1 point for each condition). The degree of gas extension into the kidney and surrounding tissues was evaluated on CT findings according to Huang–Tseng’s classification. The latter scores gas extension in four types: (i) type 1: gas limited to collecting system; (ii) type 2: gas in renal parenchyma without extension to extrarenal tissue; (iii) type 3: gas extension or abscess to perinephric (3A) or paranephric (3B) tissue; and (iv) type 4: bilateral EPN or solitary kidney with EPN. In addition, gas extension in renal parenchyma was also divided into two groups: (i) gas extension affecting < 50%; (ii) gas extension with >50% of renal parenchyma damage. Ethics committee approval was obtained by the leading center (Hospital Israelita Albert Einstein, Sao Paulo-SP/Brazil number: 5.192.573) and each center acquired its ethics board approval. All patients signed an informed consent to collect their anonymized data.

### Statistical Analysis

Categorical variables were described as frequencies and percentages. Continuous variables were described using median and interquartile ranges. The Mann–Whitney U-test was used to assess the difference between the two groups for continuous variables, whereas the chi-square test or Fisher’s exact test for categorical variables. Univariable analysis was performed based on clinical, sociodemographic, biochemical, microbiological, and radiological variables to determine the presence of ESBL pathogens. Variables significantly associated with ESBL at univariable analysis were further analyzed in multivariable analysis to identify independent factors related to the isolation of ESBL pathogens. Multivariable analysis was performed with logistic regression to find the independent variables associated with the isolation of ESBL pathogens.

A comparative analysis was performed evaluating mortality in ESBL-producing EPN and non-ESBL between the different treatment modalities: conservative, MIM, early nephrectomy, and delayed nephrectomy. Analysis was performed using SPSS for Windows, version 20.0 (IBM Corp. Armonk, NY, USA). Statistical significance was set at *p* < 0.05.

## 3. Results

A total of 570 patients met the inclusion criteria and were included for analysis. Table 1 shows patients’ clinical characteristics. Median (IQR) age was 57 (47–65) years. There were 395 women (69.3%). In 43 patients (7.5%), EPN was bilateral, and 17 patients had a solitary kidney (3.0%). More than half of the patients (58.8%) were febrile at presentation and 96 patients (16.8%) were in shock. A total of 109 patients (19.1%) had a qSOFA score ≥2 points. The most frequent symptom was flank pain (67.7%), whereas the most common biochemical alteration was leukocytosis (72.1%), followed by elevated serum creatinine (60.7%) and hyperglycemia (50.5%). Diabetes mellitus was the most frequent comorbidity (70.0%), followed by urolithiasis (52.6%). Conservative management was implemented in 66 (11.6%), early nephrectomy in 77 (13.5%), and delayed nephrectomy in 92 (16.1%). MIM was selected in 335 cases (58.7), of which 146 (25.6%) were treated with ureteral stent, percutaneous drainage in 174 (30.5%), and 15 (2.7%) with ureteral stent plus percutaneous drainage.

Table 2 shows isolated pathogens globally and by country. Among urine cultures, the most common isolated pathogen was Escherichia coli (62.2%), followed by Klebsiella spp. (20.9%). Urine culture did not isolate any pathogen in 12/556 (2.2%) patients. ESBL-producing agents were present in 291/556 urine cultures of the isolated pathogens (52.3%). India (66%), Mexico (65.9%), and Brazil (50%) were the countries with the highest rate of ESBL-producing agents.

Table 3 shows patients’ characteristics according to the presence or absence of ESBL-producing pathogens. The presence of ESBL pathogens was significantly associated at univariable analysis with thrombocytopenia (OR 1.888 95% CI 1.290–2.762, *p* = 0.001), increased serum creatinine (OR 1.884 95% CI 1.341–2.649, *p* < 0.0001), chronic kidney disease (OR 1.597 95% CI 1.131–2.255, *p* = 0.008), delayed nephrectomy (OR 2.005 95% CI 1.259–3.192, *p* = 0.003), and Huang–Tseng Scale type 4 (OR 2.672 95% CI 1.408–5.072, *p* = 0.002) (Table 3). No significant difference was observed in mortality between ESBL (13.7%) and non-ESBL (10.4%) EPN (*p* = 0.22). The qSOFA score did not show association with ESBL-producing EPN. Thrombocytopenia (OR 1.616 95% CI 1.081–2.413, *p* = 0.019) and Huang–Tseng type 4 (OR 1.948 95% CI 1.005–3.778, *p*= 0.048) were independent predictors of ESBL pathogens in multivariable analysis. Conversely, patients with Huang–Tseng Scale type 1 had 55% less chance of having ESBL-producing pathogens (OR 0.543 95% CI 0.369–0.798, *p* = 0.002) (Table 3).

Table 4 shows comparison of mortality between treatment modalities in ESBL and non-ESBL EPN. Early nephrectomy was significantly associated with increased mortality in both ESBL- (OR 2.341 95% CI 1.073–5.109, *p* = 0.029) and non-ESBL-producing infections (OR 3.760 95% CI 1.498–9.438, *p* = 0.0076). In the ESBL group, delayed nephrectomy was also significantly associated with mortality (OR 2.4 95% CI 1.163–4.950, *p* = 0.015). On the contrary, MIM reported an inverse association with mortality (OR 0.314 95% CI 0.154–0.637, *p* = 0.001).

Table 5 shows treatment modalities according to the Huang–Tseng Scale and stratified by the presence of ESBL pathogens. In type 1 EPN, a significantly greater number of patients with non-ESBL-producing pathogens were treated conservatively compared with patients harboring ESBL-producing pathogens (78.3% vs. 45.0%, *p* = 0.008). In type 4 EPN, there were significantly more patients with non-ESBL-producing pathogens requiring a minimally invasive therapy (19.2 vs. 15.8%, *p* = 0.002). Finally, there was no difference in early and delayed nephrectomy between patients with ESBL- and non-ESBL-producing pathogens.

## 4. Discussion

EPN is a life-threatening infection of the kidney and surrounding tissues caused by gas-forming organisms. Despite being a rare disease in developed countries, this condition appear to be geographically more common in Asia due to its high mortality rate and costs to healthcare systems [6].

To our best knowledge, the present study represents the largest cohort evaluating the outcomes of EPN caused by ESBL-producing agents. β-Lactamases are a cluster of enzymes present in some bacterial species and are responsible for hydrolyzing and disabling the β-lactam ring of antibiotics. Both Gram-positive and Gram-negative bacteria can produce these enzymes, but the presence of β-lactamases is one of the principal mechanisms of resistance to β-lactams in Gram-negative bacteria and particularly in Enterobacteriaceae [11].

*Escherichia coli* is the most common isolated organism in EPN, being responsible for more than 70% of infections [12], similar to our results (62.2%). Krishnamoorthy et al. reported that *Escherichia coli* was also the most common organism seen in blood cultures in patients with EPN that develop septicemia [12]. In the last decade, there has been an increased incidence of ESBL-producing agents in urinary tract infections from 21.5% to more than 40% [4,13]. In patients with EPN, a higher frequency of ESBL agents has been described compared to non-EPN urinary tract infections. Robles-Torres et al. reported ESBL agents in 31.7% of urine cultures in patients with EPN [9], compared to 21.5% in non-EPN infections in the same center [4]. In our study, an alarming 52.3% of urine cultures were positive for ESBL agents, of which 74.9% were ESBL *Escherichia coli* and 25.1% were ESBL Klebsiella spp.

Over the years, there was a trend towards an increase in ESBL-producing pathogens in urinary tract infections and especially in EPN [13]. Risk factors have been described for the presence of ESBL agents in urinary tract infections: prior antibiotic use, previous hospitalizations, chronic corticosteroids use, invasive procedures (indwelling catheters, gastrostomy, nasogastric tube, hemodialysis, arterial pathways), poor nutritional status, advanced age, and diabetes mellitus [14]. The most common antibiotics associated with ESBL agents are prior use of third-generation cephalosporins and quinolones. Other risk factors described include recurrent urinary tract infections, high comorbidity (>2 points in Charlson Index), immunocompromised status, urolithiasis, and complicated urinary tract infections (anatomical or functional abnormalities in the urinary tract). ESBL-producing infections were also associated with worse symptoms and longer hospital stay [4,15]. However, we found no association of ESBL-producing EPN with age, diabetes, longer hospital stays, severity of symptoms, or urolithiasis, but we found in multivariable analysis that the risk of ESBL pathogen EPN was almost two-fold higher in patients with Huang–Tseng type 4, whereas patients with type 1 had 55% less chance of having ESBL-producing pathogens.

Inability to identify patients at risk of ESBL-producing infections and starting inappropriate empirical antibiotic therapy may worsen the prognosis and potentially increase mortality in EPN, which is already a life-threatening infection. Despite this, mortality has decreased in the last decades due to the introduction of better imaging studies and minimally invasive therapies, and the mortality rate currently ranges from 6–20.6% [12,16,17]. Our study showed that overall mortality was 12.1%. Interestingly, we found that mortality did not differ significantly between ESBL and non-ESBL infections. However, sub-analysis according to the type of treatment showed that mortality in the ESBL group was significantly higher in patients who underwent both early and delayed nephrectomy, and patients treated with MIM were demonstrated to have lower odds of mortality. In the non-ESBL group, patients who underwent early nephrectomy also demonstrated a significantly higher mortality rate. These results could be explained by the fact that patients who had early nephrectomy were more gravely ill (e.g., hemodynamic instability) and, as a consequence, had a higher mortality because of their presenting medical condition. In addition, the reduced glomerular filtration after nephrectomy may also have contributed to mortality. Therefore, patients with an ESBL infection should be probably managed with antibiotics, supportive therapy, and MIM as much as possible, since mortality was also significantly higher in those who underwent delayed nephrectomy.

Few other studies have described the microbiological characteristics and their association with prognosis in patients with EPN. Lu et al. reported that third-generation cephalosporin resistance, polymicrobial infections, and previous antibiotic use were risk factors for increased mortality [18]. In addition, they concluded that prior hospitalization, prior antibiotic use, need for hemodialysis, and disseminated intravascular coagulation were factors associated with third-generation cephalosporin-resistant uropathogens. In patients with these risk factors for antibiotic resistance, carbapenem is the empiric antibiotic of choice [18]. Jain et al. described a mortality scoring system for EPN based on several risk factors, including multidrug-resistant uropathogens [19]. They reported a 45% resistance to third-generation cephalosporins. Based on these results, they recommended initiating carbapenem when a patient has already been on a cephalosporin medication.

In a retrospective study of 63 patients with EPN, Arrambide-Herrera et al. described that ESBL-producing bacteria and multidrug-resistant bacteria (including ESBL plus resistance to trimethoprim-sulfamethoxazole and quinolones) were not associated with increased mortality and intensive care unit admission [16]. In a recent study, ESBL-producing agents were associated with leukocytosis >11,000/mL in univariable but not in multivariable analysis. Prognostic outcomes such as qSOFA score, Huang–Tseng classification, intensive care unit admission, and mortality were not associated with ESBL-producing organisms [9]. The authors also reported resistance profiles and showed that antibiotic resistance was high for levofloxacin (50%), ciprofloxacin (63.1%), and trimethoprim-sulfamethoxazole (87%). Colistin (4.4%), meropenem (8.7%), and Fosfomycin (19.5%) were the antibiotics with less resistance reported. [8]

In our study, multivariable analysis reported that thrombocytopenia and Huang type 4 were associated with ESBL-producing agents. Most patients with Huang type 4 EPN reported impaired kidney function, probably affecting the concentration of the antibiotics in the renal parenchyma and urinary tract. Similar to the results reported by Robles-Torres, ESBL agents were not associated with the severity of the disease evaluated by qSOFA score and mortality [9]. However, in our sub-analysis, patients with ESBL infections were significantly associated with mortality after nephrectomy, whereas MIM significantly reduced the risk of death.

Regarding correlation between treatments and EPN diffusion in ESBL and non-ESBL patients, we found that there was a significant difference in treatments between Huang–Tseng Scale type 1. A greater number of non-ESBL type 1 patients were treated conservatively compared with ESBL. This may be because ESBL patients presented with a more severe clinical condition.

Our study presents some limitations, beginning with its retrospective nature. Different therapeutic algorithms between centers were implemented, influenced by the lack of standardized recommendations in the literature. Furthermore, we were unable to report antibiotic therapy implemented in most of the centers, resulting in limited data in order to propose antibiotic management protocols. Some well-known risk factors for ESBL agents were not reported, including recurrent urinary tract infections and previous antibiotic use. Despite reporting the uropathogens in the analyzed urine cultures, we were unable to obtain the complete resistance profile in most of the included centers.

This study provides a basis for further prospective studies evaluating different antibiotic protocols in order to improve outcomes in EPN patients.

## 5. Conclusions

In this large, multicenter study, we analyzed the mortality of EPN patients according to the presence or not of ESBL-producing agents. ESBL-producing pathogens were isolated in more than half of urine cultures (52.3%) We found that the mortality rate was not significantly higher in patients with ESBL-producing pathogens as compared to those without. Patients with ESBL infection demonstrated to have a poor prognosis when treated with early or delayed nephrectomy, whereas the mortality was significantly lower in those patients treated with MIM.

## Figures and Tables

**Table 1 pathogens-11-01397-t001:** Demographic, clinical, biochemical, and radiological characteristics of the study population. (n = 570).

Variables	Median (IQR) or n (%)
Demographics	
Age (years); median (IQR)	57 (47–65)
Female	395 (69.3)
Right kidney	242 (42.5)
Left kidney	268 (47)
Bilateral	43 (7.5)
Solitary kidney	17 (3)
Days of hospital stay; median (IQR)	9 (6–14)
Clinical characteristics	
Hypotension (BP < 90/60 or MAP < 60 mmHg)	96 (16.8)
Fever (>38.3 °C)	335 (58.8)
Flank pain	386 (67.7)
Lower urinary tract symptoms	120 (21.1)
Death	69 (12.1)
qSOFA score	
0 points	306 (53.7)
1 point	155 (27.2)
2 points	73 (12.8)
3 points	36 (6.3)
Biochemical characteristics	
Anemia (Hb <12 g/dL)	227 (39.8)
Leukocytosis (>11,000/μL)	411 (72.1)
Leukopenia (<4500/μL)	14 (2.5)
Thrombocytopenia (<150,000/μL)	152 (26.7)
Hyperglycemia (glucose >200 mg/dL)	288 (50.5)
Increased creatinine (serum Cr >1.2 mg/dL)	346 (60.7)
Comorbidities	
Diabetes mellitus	399 (70)
Urolithiasis	300 (52.6)
Chronic kidney disease	207 (36.3)
Neurogenic bladder	27 (4.7)
Oncologic disease	17 (3.0)
Antibiotic resistance	
ESBL agents (n = 556) #	291 (52.3)
Huang–Tseng Scale	
Type 1	168 (29.5)
Type 2	144 (25.3)
Type 3a	109 (19.1)
Type 3b	99 (17.4)
Type 4	50 (8.7)
Renal parenchyma extension	
Gas extension > 50%	109 (19.1)
Management	
Conservative	66 (11.6)
Early nephrectomy	77 (13.5)
Ureteral stent	146 (25.6)
Percutaneous drainage	174 (30.5)
Ureteral stent + percutaneous drainage	15 (2.7)
Delayed nephrectomy *	92 (16.1)

* Patients refractory to minimally invasive management. IQR = interquartile range; BP = blood pressure; MAP = mean blood pressure; Hb = hemoglobin; Cr = creatinine; ESBL = extended-spectrum beta-lactamase. # 14 urine cultures were missing due to early mortality.

**Table 2 pathogens-11-01397-t002:** Microbiological profile of urine cultures pathogens by country. (n = 556 *).

Microbiological Agents	Total; n (%)	Mexico (n = 232)	India (n = 100)	Arabia Saudi (n = 56)	Turkey (n = 32)	Malaysia (n = 30)	Nepal (n = 26)	Brazil (n = 20)	Singapore (n = 20)	Morocco (n = 18)	Hong Kong (n = 14)	Burkina Faso (n = 8)
*E. coli*	346 (62.2)	160 (69)	60 (60)	29 (51.8)	18 (56.3)	14 (46.7)	11 (42.3)	15 (75)	13 (65)	12 (66.7)	11 (78.6)	3 (37.5)
*Klebsiella* spp.	116 (20.9)	43 (18.5)	16 (16)	20 (35.7)	8 (25)	8 (26.7)	8 (30.8)	1 (5)	5 (25)	3 (16.7)	2 (14.3)	2 (25)
*Proteus mirabilis*	21 (3.8)	2 (0.9)	7 (7)	3 (5.4)	0	1 (3.3)	5 (19.2)	2 (10)	0	0	0	1 (12.5)
*Candida* spp.	15 (2.7)	10 (4.3)	1 (1)	2 (3.6)	1 (3.1)	0	0	0	1 (5)	0	0	0
*Pseudomonas aeruginosa*	15 (2.7)	1 (0.4)	7 (7)	0	1 (3.1)	2 (6.6)	1 (3.8)	0	0	1 (5.6)	1 (7.1)	1 (12.5)
*Morganella morganii*	9 (1.6)	1 (0.4)	1 (1)	2 (3.6)	0	3 (10)	1 (3.8)	0	1 (5)	0	0	0
*Enterococcus faecalis*	8 (1.4)	2 (0.9)	4 (4)	0	1 (3.1)	0	0	1 (5)	0	0	0	0
Negative cultures	14 (2.5)	10 (4.3)	2 (2)	0	1 (3.1)	0	0	0	0	1 (5.6)	0	0
Others ^2^	12 (2.2)	3 (1.3)	2 (2)	0	2 (6.3)	2 (6.6)	0	1 (5)	0	1 (5.6)	0	1 (12.5)
ESBL-producing agents	291 (52.3)	153 (65.9)	66 (66)	13 (23.2)	15 (46.8)	14 (46.6)	6 (23.1)	10 (50)	6 (30)	1 (5.6)	6 (42.8)	1 (12.5)
ESBL *E. coli* ^1^	218 (74.9)	119 (77.8)	52 (78.8)	7 (53.8)	10 (66.7)	9 (64.3)	2 (33.3)	9 (90)	4 (66.6)	0	5 (83.3)	1 (100)
ESBL *Klebsiella* spp. ^1^	73 (25.1)	34 (22.2)	14 (21.2)	6 (35.7)	5 (33.3)	5(35.7)	4 (66.7)	1 (10)	2 (33.3)	1 (100)	1 (16.7)	0

* 14 urine cultures were missing due to early mortality. ^1^ Percentage among ESBL agents. ^2^ Enterobacter cloacae n = 3 (0.5%), Staphylococcus aureus n = 6 (1.1%), Acinetobacter baumannii n = 3 (0.5%).

**Table 3 pathogens-11-01397-t003:** Univariate and multivariate analysis for clinical, biochemical, and radiological factors associated with ESBL-producing agents in patients with emphysematous pyelonephritis. (n = 570).

Variables	ESBL (n = 291)	Non-ESBL (n = 279)	Univariate		Multivariate	
			*p* Value	OR (IC 95%)	*p* Value	OR (IC 95%)
Sociodemographics						
Female	195 (67)	200 (71.7)	0.226	0.802 (0.561–1.147)		
Age (years); median (IQR)	57 (46–65)	57 (47–64)	0.333 *	-		
Days of hospital stay; median (IQR)	9 (6–14)	9 (6–14)	0.935 *	-		
Clinical characteristics						
Fever (>38.3 °C)	165 (56.7)	170 (60.9)	0.305	0.840 (0.601–1.173)		
Flank pain	195 (67)	191 (68.5)	0.712	0.936 (0.659–1.330)		
Shock (BP < 90/60 or MAP < 60 mmHg)	45 (15.5)	51 (18.3)	0.369	0.818 (0.527–1.269)		
Mortality	40 (13.7)	29 (10.4)	0.22	1.374 (0.826–2.286)		
qSOFA score						
0 pts	148 (50.9)	158 (56.6)	0.167	0.793 (0.570–1.102)		
1 pt	86 (29.6)	69 (24.7)	0.196	1.277 (0.881–1.850)		
2 pts	40 (13.7)	33 (11.8)	0.493	1.188 (0.725–1.946)		
3 pts	17 (5.8)	19 (6.8)	0.635	0.849 (0.432–1.669)		
Biochemical characteristics						
Anemia (Hb < 12 g/dL)	125 (43)	102 (36.6)	0.119	1.307 (0.933–1.829)		
Leukocytosis (>11,000/μL)	204 (70.1)	207 (74.2)	0.276	0.816 (0.565–1.178)		
Thrombocytopenia (<150,000/μL)	95 (32.6)	57 (20.4)	0.001	1.888 (1.290–2.762)	0.019	1.616 (1.081–2.413)
Hyperglycemia (glucose > 200 mg/dL)	139 (47.8)	149 (53.4)	0.178	0.798 (0.574–1.109)		
Increased creatinine (serum Cr > 1.2 mg/dL)	198 (68)	148 (53)	<0.001	1.884 (1.341–2.649)	0.108	1.382 (0.932–2.049)
Comorbidities						
Diabetes mellitus	202 (69.4)	197 (70.6)	0.756	0.945 (0.660–1.352)		
Chronic kidney disease	121 (41.6)	86 (20.8)	0.008	1.597 (1.131–2.255)	0.244	1.263 (0.853–1.870)
Urolithiasis	158 (54.3)	142 (50.9)	0.416	1.146 (0.825–1.593)		
Neurogenic bladder	9 (3.1)	18 (6.5)	0.059	0.463 (0.204–1.048)		
Huang–Tseng Scale						
Type 1	62 (21.3)	106 (38)	<0.001	0.442 (0.305–0.640)	0.002	0.543 (0.369–0.798)
Type 2	74 (25.4)	70 (25.1)	0.926	1.018 (0.698–1.486)		
Type 3 a	64 (22)	45 (16.1)	0.075	1.466 (0.961–2.237)		
Type 3 b	55 (18.9)	44 (15.8)	0.324	1.245 (0.805–1.924)		
Type 4	33 (12.4)	14 (5)	0.002	2.672 (1.408–5.072)	0.048	1.948 (1.005–3.778)
Renal parenchyma extension						
Gas extension > 50%	64 (22)	45 (16.1)	0.075	1.466 (0.961–2.237)		

* Mann–Whitney U-Test; OR = odds ratio; CI = confidence interval. Hosmer–Lemeshow test of 0.688.

**Table 4 pathogens-11-01397-t004:** Mortality among ESBL- and non-ESBL-producing emphysematous pyelonephritis stratified by treatment modality.

	ESBL				Non-ESBL			
Management	Mortality (n = 40)	Non-Mortality (n = 251)	*p* Value	OR (IC 95%)	Mortality (n = 29)	Non-Mortality (n = 250)	*p* Value	OR (IC 95%)
Conservative	2 (5)	18 (7.2)	0.999	0.681 (0.152–3.055)	3 (10.3)	43 (17.2)	0.437 *	0.555 (0.161–1.918)
Minimally invasive therapy	13 (32.5)	152 (60.6)	0.001	0.314 (0.154–0.637)	14 (48.3)	156 (62.4)	0.14	0.562 (0.260–1.217)
Early nephrectomy	11 (27.5)	35 (13.9)	0.029	2.341 (1.073–5.109)	8 (27.6)	23 (9.2)	0.0076 *	3.760 (1.498–9.438)
Delayed nephrectomy	14 (35)	46 (18.3)	0.015	2.4 (1.163–4.950)	4 (13.8)	28 (11.2)	0.757 *	1.269 (0.411–3.913)

* Fisher’s exact test.

**Table 5 pathogens-11-01397-t005:** Stratification of patients according to Huang–Tseng classification and treatment modality.

	Conservative			Minimally Invasive Therapy			Early Nephrectomy			Delayed Nephrectomy		
Huang–Tseng Scale	ESBL (n = 20)	Non-ESBL (n = 46)	*p* Value	ESBL (n = 165)	Non-ESBL (n = 170)	*p* Value	ESBL (n = 46)	Non-ESBL (n = 31)	*p* Value	ESBL (n = 60)	Non-ESBL (n = 32)	*p* Value
Type 1	9 (45)	36 (78.3)	**0.008**	47 (28.5)	63 (39.8)	0.095	2 (4.3)	3 (9.7)	0.387 *	4 (6.7)	4 (6.7)	0.442 *
Type 2	6 (30)	6 (13)	0.101	46 (27.9)	48 (33.8)	0.942	8 (17.4)	9 (29)	0.227	14 (23.3)	7 21.9)	0.874
Type 3A	2 (10)	1 (2.2)	0.216 *	32 (19.4)	26 (22.4)	0.321	17 (37)	9 (29)	0.471	13 (21.7)	9 (28.1)	0.489
Type 3B	0 (0)	1 (2.2)	0.999 *	14 (8.5)	24 (24)	0.104	18 (39.1)	10 (32.3)	0.539	23 (38.3)	9 (28.1)	0.328
Type 4	3 (15)	2 (4.3)	0.159 *	26 (15.8)	9 (19.2)	**0.002**	1 (2.2)	0 (0)	0.999 *	6 (10)	3 (9.4)	0.999 *

* Fisher’s exact test. Bold value stands for significant *p* value

## Data Availability

Data will be provided by the corresponding author by a reasonable request.

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
