# Peer review of "Prognosis of Extended-Spectrum-Beta-Lactamase-Producing Agents in Emphysematous Pyelonephritis-Results from a Large, Multicenter Series"

_pathogens, 2022, doi:10.3390/pathogens11121397_

Round 1

Reviewer 1 Report

Prognosis of extended spectrum beta-lactamase producing agents in emphysematous pyelonephritis. Results from a large, 3 multicenter series. José Iván Robles-Torres et al.

Thank you to the authors for their submission to Pathogens. This study is a retrospective review of 22 centers’ databases concerning patients with emphysematous pyelonephritis (EPN). This pathology is rare and severe.  This survey is probably one of the most important by the number of patients included. The method is in agreement with the title which is that of a descriptive study, about the prognosis of patients with ESBL agents in emphysematous pyelonephritis. However the aim of the study about the risk factors must be clarified. If the authors wanted to determine whether ESBL is an aggravating factor in emphysematous pyelonephritis, they should have compared pyelonephritis due to the same bacterial species, for example ESBL E. coli vs non-producing ESBL.

Statistical tests help to confirm the interpretation of the results.  Use of Huang-Tseng type 1 as a reference and not include it in multivariate analysis is questionable (see below).

Detailed comments:

Page 2 line 52 “Escherichia coli was the most frequent pathogen isolated (62.4%).” This sentence formulation is ambiguous. The reader might understand that among the isolated pathogen there were 62.4% E. coli, when in fact 62.4% of urine cultures yielded E. coli.

Page 2 line 75 the authors must provide one or two other references to assess that “Infections caused by these organisms (Klebsiella spp and Proteus spp) are associated with a “high mortality rate” What do they mean by high?

Page 2 line 78-76 some authors used imipenem in the EPN treatment.

Page 3 line 119, in which country the Ethics committee approval was obtained?

Page 3 table 1

Why for the Huang-Tseng scale are there 572 patients with ESBL instead of 570?

Page 5 lines 154-155: The sentence formulation for E. coli and Klebsiella percentage is ambiguous. (see Page 2 line 52 comment).

Page 5 line 155: the authors mentioned a total of 556 urine cultures. Then the total number of E. coli is 346 and not 356 as mentioned in table 2. They assess that ESBL were present in 291 urine cultures such a presentation is receivable. But the following sentence, lines 157 and 158, is questionable because usually the rate of ESBL-producing agents are reported to a number of bacteria able to produce ESBL. The percentages given for E. coli or Klebsiella spp are more pertinent.

Page 5 lines 157 the percentage of ESBL producing agents in Mexico is 65.9 instead of 61.4

Table 2 the percentage of Klebsiella spp in Arabia Saudia is 35.7 instead of 25.7

the percentage of ESBL producing agents in Mexico is 65.9 instead of 61.4

the percentage of ESBL Klebsiella spp in Nepal is 66.7 instead of 66.4, and in Singapure, 33.3 instead of 33.7.

Page 5 lines 167: Huang-Tseng Type 2 and 3a percentages don’t differ statistically between ESBL and non ESBL. If p < 0.05 it is because Type 1 percentages are different between ESBL and non-ESBL (21.3% vs 38%; p=0.00001, OR 0.44).  It is not correct to use Type 1 value as a reference and to exclude the type 1 from the multivariate analysis. It is a misuse of the test, and a falsification of the results ‘presentation.

Table 3

I don’t agree with the use of the comparison of each Huang-Tseng Type to Type 1 because the p value is more linked to the distribution difference between ESBL and Non-ESBL for type 1 than for other types.  The aim of the statistical test is to check whether the observed difference is due to chance or not. For the type 2 especially there is no difference (25.4% vs 25.1%). The difference observed for the types 3a and 3b are not statistically significant when they are tested respectively versus non-type3a and non-type 3b. The only differences statistically significant are for type 1 and type 4. This comment is applicable for the qSOFA analysis.

Why for the Huang-Tseng scale are there 293 patients with ESBL instead of 291?

Table 4

For conservative treatment in ESBL group the p value was obtained with the Fisher exact test. This must be mentioned.

For Early Nephrectomy in non ESBL group, the Fisher exact test must be used. The p value is 0.0076 instead of 0.003

For Delayed Nephrectomy in non ESBL group the p value was obtained with from the Fisher exact test. This must be mentioned.

Page 7 line 189 Ubee, S.S. et al. did not exactly write that it was “a serious problem in developing countries” but that these infections “appear to be geographically more common in Asia”.

Page 7 line 195: to my knowledge there is no ESBL in Gram positive bacteria. The authors must add a reference showing it.

Page 8 lines 231 to 242, it would be interesting to have data about the association between Huang scale and treatment.

Page 8 lines 267 Huang type 2 was not associated with ESBL-producing agents. (see Page 5 lines 167 comment)

References

Page 10 line 336, reference 8: year and details of publication reference must be corrected (2022)

Typing mistakes:

Page 1 line 47 « Cemographics »

Page 1 line 52 62.4% E. coli instead of 62.2% in the table 2

P2 Line 72 « Klebsiella spp and Proteus spp” italicize

Page 3 line 133: “p<0.05.3. »

Page 4 Table 1 Fever > 38°3 °C instead “<38°3 °C”

qSOFA score : 2 points

Table 2: Morganella morganii

Table 3: “Hyperglycemia”

Page 7 line 209: italicize  “Klebsiella spp”

Author Response

Thank you to the authors for their submission to Pathogens. This study is a retrospective review of 22 centers’ databases concerning patients with emphysematous pyelonephritis (EPN). This pathology is rare and severe.  This survey is probably one of the most important by the number of patients included. The method is in agreement with the title which is that of a descriptive study, about the prognosis of patients with ESBL agents in emphysematous pyelonephritis. However the aim of the study about the risk factors must be clarified. If the authors wanted to determine whether ESBL is an aggravating factor in emphysematous pyelonephritis, they should have compared pyelonephritis due to the same bacterial species, for example ESBL E. coli vs non-producing ESBL.

REPLY. We would like to thank you for your nice words on our study. We would like to highlight that this study's focus is not analyzing the aggravating factors of EPN but instead focuses on understanding the prognostic variables that define ESBL infections  in patients  diagnosed with EPN. Therefore, the study aim was rephrased in introduction as follows “The aim of this study was to  determine the  clinical factors and their influence on  prognosis in patients being managed for EPN  with and without  ESBL-producing bacteria and to identify if those with EPN due to ESBL infections fared any different.

Statistical tests help to confirm the interpretation of the results.  Use of Huang-Tseng type 1 as a reference and not include it in multivariate analysis is questionable (see below).

REPLY. We would like to thank you for this comment. A multivariable logistic regression analysis dealing with a categorical variable always needs a reference to be considered for comparison. That is why we used Huang-Tseng type 1 as the reference. Considering that Huang-Test variable is a 5 elements variable, a reference must be included for analysis. Analyzing this variable as dichotomous (i.e. yes/no) would be methodologically wrong.

Detailed comments:

Page 2 line 52 “Escherichia coli was the most frequent pathogen isolated (62.4%).” This sentence formulation is ambiguous. The reader might understand that among the isolated pathogen there were 62.4% E. coli, when in fact 62.4% of urine cultures yielded E. coli.

REPLY. We would like to thank you for this comment. We do apologize for not being clear. we rephrased that sentence as follows “Among positive cultures, the most common isolated pathogen was Escherichia coli (62.2%).”

Page 2 line 75 the authors must provide one or two other references to assess that “Infections caused by these organisms (Klebsiella spp and Proteus spp) are associated with a “high mortality rate” What do they mean by high?

REPLY. We would like to thank you for this comment. We do apologize for not being clear. we mean that Infections caused by ESBL producing pathogens are associated with higher mortality than corresponding infections due to non-ESBL pathogens. We rephrased that sentence as “Infections produced by ESBL producing pathogens are associated with higher mortality than corresponding infections due to non-ESBL pathogens, with reported mortality rate between 3.7 and 22.1%”. We also provided more references supporting this (i.e.: PMID: 30986558; PMID: 30208454)

Page 2 line 78-76 some authors used imipenem in the EPN treatment.

REPLY. We would like to thank you for this comment. We do agree with you but we are referring in the text to what cited reference says.

Page 3 line 119, in which country the Ethics committee approval was obtained?

REPLY. We would like to thank you for this comment. We explained that as follows in the main text “Ethics committee approval was obtained by the leading center (Hospital Israelita Albert Einstein,Sao Paulo-SP/ Brazil number: 5.192.573) and each center acquired its ethics board approval.

Page 3 table 1

Why for the Huang-Tseng scale are there 572 patients with ESBL instead of 570?

REPLY. We would like to thank you for this comment. We do apologize for that mistake. We double checked data in Table 1 and there was a mistake in Huang-Tseng scale type 1 where there were 166 patients instead of 168. Both Tables 1 and 3 were amended accordingly. Thank you very much for pointing out this

Page 5 lines 154-155: The sentence formulation for E. coli and Klebsiella percentage is ambiguous. (see Page 2 line 52 comment).

REPLY. We would like to thank you for this comment. We do apologize for not being clear. we rephrased that sentence as follows “Among positive cultures, the most common isolated pathogen was Escherichia coli (62.2%), followed by Klebsiella spp. (20.9%).”

Page 5 line 155: the authors mentioned a total of 556 urine cultures. Then the total number of E. coli is 346 and not 356 as mentioned in table 2. They assess that ESBL were present in 291 urine cultures such a presentation is receivable. But the following sentence, lines 157 and 158, is questionable because usually the rate of ESBL-producing agents are reported to a number of bacteria able to produce ESBL. The percentages given for E. coli or Klebsiella spp are more pertinent.

Page 5 lines 157 the percentage of ESBL producing agents in Mexico is 65.9 instead of 61.4

REPLY. We would like to thank you for this comment. We do apologize for this mistake that was amended.

Table 2 the percentage of Klebsiella spp in Arabia Saudia is 35.7 instead of 25.7

REPLY. We would like to thank you for this comment. We do apologize for this mistake that was amended.

the percentage of ESBL producing agents in Mexico is 65.9 instead of 61.4

REPLY. We would like to thank you for this comment. We do apologize for this mistake that was amended.

the percentage of ESBL Klebsiella spp in Nepal is 66.7 instead of 66.4, and in Singapore, 33.3 instead of 33.7.

REPLY. We would like to thank you for this comment. We do apologize for this mistake that was amended.

Page 5 lines 167: Huang-Tseng Type 2 and 3a percentages don’t differ statistically between ESBL and non ESBL. If p < 0.05 it is because Type 1 percentages are different between ESBL and non-ESBL (21.3% vs 38%; p=0.00001, OR 0.44).  It is not correct to use Type 1 value as a reference and to exclude the type 1 from the multivariate analysis. It is a misuse of the test, and a falsification of the results ‘presentation.

REPLY. We would like to thank you for this comment. A multivariable logistic regression analysis dealing with a categorical variable always needs a reference to be considered for comparison. That is why we used Huang-Tseng type 1 as the reference. Analyzing this variable as dichotomous (i.e. yes/no) would be methodologically wrong.

Table 3

I don’t agree with the use of the comparison of each Huang-Tseng Type to Type 1 because the p value is more linked to the distribution difference between ESBL and Non-ESBL for type 1 than for other types.  The aim of the statistical test is to check whether the observed difference is due to chance or not. For the type 2 especially there is no difference (25.4% vs 25.1%). The difference observed for the types 3a and 3b are not statistically significant when they are tested respectively versus non-type3a and non-type 3b. The only differences statistically significant are for type 1 and type 4. This comment is applicable for the qSOFA analysis.

REPLY. We would like to thank you for this comment. The results reported in Table 3 are those retrieved from the statistical software even if they are similar.

Why for the Huang-Tseng scale are there 293 patients with ESBL instead of 291?

REPLY. We would like to thank you for this comment. We do apologize for this mistake that was amended

Table 4

For conservative treatment in ESBL group the p value was obtained with the Fisher exact test. This must be mentioned.

REPLY. We would like to thank you for this comment. Statistical analysis was implemented in the main text according to your suggestion as follows “The Mann-Whitney U-test was used to assess the difference between the two groups for continuous variables, whereas the Chi-square test or Fisher exact test for categorical variables

For Early Nephrectomy in non ESBL group, the Fisher exact test must be used. The p value is 0.0076 instead of 0.003

REPLY. We would like to thank you for this comment. We amended table 4 and main test according to that comment

For Delayed Nephrectomy in non ESBL group the p value was obtained with from the Fisher exact test. This must be mentioned.

REPLY. We would like to thank you for this comment. We amended table 4 according to that comment

Page 7 line 189 Ubee, S.S. et al. did not exactly write that it was “a serious problem in developing countries” but that these infections “appear to be geographically more common in Asia”.

REPLY. We would like to thank you for this comment. We amended the text according to you suggestion

Page 7 line 195: to my knowledge there is no ESBL in Gram positive bacteria. The authors must add a reference showing it.

REPLY. We would like to thank you for this comment. We do apologize for not being clear. We did not mean that Gram positive bacteria are ESBL producing pathogens but just that they can produce β-Lactamases (see for example this paper: https://www.cureus.com/articles/90158-magnitude-of-extended-spectrum-beta-lactamase-producing-gram-negative-and-beta-lactamase-producing-gram-positive-pathogens-isolated-from-patients-in-dar-es-salaam-tanzania-a-cross-sectional-study). Indeed, we say “β-Lactamases are a cluster of enzymes present in some bacterial species and are responsible for hydrolyzing and disabling the β-lactam ring of antibiotics. Both Gram-positive and Gram-negative bacteria can produce these enzymes but the presence of β-lactamases is one of the principal mechanisms of resistance to β-lactams in Gram-negative bacteria and particularly in Enterobacteriaceae”.

Page 8 lines 231 to 242, it would be interesting to have data about the association between Huang scale and treatment.

REPLY. We would like to thank you for this important comment. We performed stratification of patients considering the Huang-Tseng scale and treatment modality. Data are presented in Table 5. We found that there was a significant difference in treatments between Huang-Tseng scale Type 1 and Type 3b. A greater number of non-ESBL Type 1 patients were treated conservatively and with minimally invasive therapy compared with ESBL. This may be because ESBL patients presented with a more severe clinical condition. There were also more Type 3b non-ESBL patients treated conservatively, whereas more Type 3b ESBL patients required a delayed nephrectomy. The latter could reflect a disease not-responding to conservative and minimally invasive treatments in ESBL patients. indeed, Type 3b commonly affects a bulky part of the kidney parenchyma and spreads beyond Gerota's fascia, reaching the paranephric space and affecting the surrounding organs. In contrast, Type 4 EPN involves both kidneys or a solitary kidney, but the gas extension does not spread as largely as in Type 3b EPN. We added this in Results and discussion sections.

Page 8 lines 267 Huang type 2 was not associated with ESBL-producing agents. (see Page 5 lines 167 comment)

REPLY. We would like to thank you for this comment. Huang type 2 was associate with ESBL producing agent on both univariable and multivariable analysis (please see Table 3; multivariable OR 1.751 95% CI 1.106-2.772). As we clarified before, categorical variables (with more than 2 elements) are recommended to be tested with a reference value (like type 1 Huang).

References

Page 10 line 336, reference 8: year and details of publication reference must be corrected (2022)

REPLY. We would like to thank you for this comment. We do apologize for this mistake. That reference was amended

Typing mistakes:

REPLY. We would like to thank you for pointing out this. We do apologize for these mistakes which were all amended

Page 1 line 47 « Cemographics »

Page 1 line 52 62.4% E. coli instead of 62.2% in the table 2

P2 Line 72 « Klebsiella spp and Proteus spp” italicize

Page 3 line 133: “p<0.05.3. »

Page 4 Table 1 Fever > 38°3 °C instead “<38°3 °C”

qSOFA score : 2 points

Table 2: Morganella morganii

Table 3: “Hyperglycemia”

Page 7 line 209: italicize  “Klebsiella spp”

Reviewer 2 Report

The manuscript of Robles-Torres et al. is interesting and can be accepted for publication after some changes.

1. The Authors wrote that one of the risk factor of ESBL isolates infection is use of 3-rd generation cephalosporins and quinolones. It would be good to add susceptibility to these antibiotics.

2. Some up-to-date references are missing (e.g. Cheng et al. 2011, Johansen & Naber 2014).

3. The Authors should specify what species of Klebsiella spp. were estimated.

4. What do you mean by term "Pseudomonas"? Pseudomonas spp. or e.g. Pseudomonas aeruginosa.

5. Some names of the bacteria are spelled wrong. 

6. As far as I know there isn' t species called Providencia baumani. The Authors should check it. 

Author Response

Review 2

The manuscript of Robles-Torres et al. is interesting and can be accepted for publication after some changes.

REPLY. We would like to thank you for your nice comments on our study

  1. The Authors wrote that one of the risk factor of ESBL isolates infection is use of 3-rd generation cephalosporins and quinolones. It would be good to add susceptibility to these antibiotics.

REPLY. We would like to thank you for this important comment. Unfortunately, we cannot provide this because database was anonymized and antibiotics susceptibility was not gathered.

  1. Some up-to-date references are missing (e.g. Cheng et al. 2011, Johansen & Naber 2014).

REPLY. We would like to thank you for suggesting us these references. Unfortunately, we were not able to find them just using first author name and year of publication. Therefore, in case you deem necessary, we kindly ask you to provide us with full length title, authors, and Journal name. Thank you

  1. The Authors should specify what species of Klebsiella spp. were estimated.

REPLY. We would like to thank you for this important comment. Unfortunately, we did not get this data and the anonymized database does not allow us to gather it

  1. What do you mean by term "Pseudomonas"? Pseudomonas spp. or e.g. Pseudomonas aeruginosa.

REPLY. We would like to thank you for this comment. It was Pseudomonas aeruginosa that was added in Table 2

  1. Some names of the bacteria are spelled wrong. 

REPLY. We would like to thank you for this comment. We do apologize for this mistake. We replaced the wrong “Staphylococcus aereus” with the correct one “Staphylococcus aureus”

  1. As far as I know there isn' t species called Providencia baumani. The Authors should check it.

REPLY. We would like to thank you for this comment. We do apologize for this mistake. We double checked and it is Acinetobacter baumannii. We changed that in table

Round 2

Reviewer 1 Report

In this second version the authors did not take my objections into account.

I don't agree the way the authors used the statistical tests for the Huang Scale score (lines 181-183).

They did not provided the tables in the revised version

Author Response

In this second version the authors did not take my objections into account. I don't agree the way the authors used the statistical tests for the Huang Scale score (lines 181-183).

REPLY: Thank you very much for your comment. We followed your suggestion and provided a new analysis according to your comment in revision round 1. We provided a new analysis inserting Huang Scale type 1. The new analysis is available in Table 3. Discussion has been changed and highlighted according to new results. Thank you again for your useful suggestion

They did not provided the tables in the revised version

REPLY: Thank you very much for your comment. Tables are provided as a .zip file and added to manuscript

Round 3

Reviewer 1 Report

The article requires minor revisions and some numbers and calculations must be checked.

Page 2 line 57 and 58 in fact 62.2% Escherichia coli was not among the positive cultures, but among the urine cultures. It must be corrected.

Pages 2 lines 58 to 61 the results must be corrected according to Huang-Tseng scale results of the table 3 and of the page 6 line 170.

Page 4 line 66: 62.2% Escherichia coli was not among the positive cultures, but among the urine cultures. This must be corrected.

Page 4 and 5 Table 1: the number of Type1 Huang Scale must be corrected: 166 (29.1%) instead of 168, as the authors said in their answer (166 instead of 168 for type 1)..

Page 5 Table 2:

the total number of E. coli is 346 and not 356 (62.2% of 556 urine cultures; with 356 E. coli the total number of urine cultures is 566).

The percentage of Klebsiella spp in Arabia Saudia is 35.7 instead of 25.7

The percentage of ESBL agents in Mexico is 65.9% instead of 66.6%

Page 5 lines 159-160 The percentage of E.coli and K.pneumoniae are not reported to the positive cultures but to the total of urine cultures (negative included). It must be corrected.

Page 6 Table 3: Fever > 38.3°C instead of “<”

The number of Type1 Huang Scale in the ESBL group must be corrected: 60 (20.6%) instead of 62

Page 7 Lines 187-192 and table 5

I don’t understand how were the percentages calculated. For example, line 188, in Type 1 EPN there are 61 patients in the ESBL group, 9 of them received a conservative treatment. 9/61 = 14.8% instead of 5.4%. All the percentage of table 5 must be changed.

Page 7 table 5

The total number of patients with ESBL is 290 instead of 291 in the table 3 and the number of patients without ESBL is 280 instead of 279 in the table 3.

The total number of patients with Huang Scale: type2 is 145 instead of 144 in table 1, type 3a 107 instead of 109 in table 1, and type 3b 100 instead of 99 in table 1.

Some p values must be checked: 0.44 instead of 0.88 for type 3B conservative, 0.56 instead of 0.9 for type 4 conservative, 0.74 instead of 0.64 for type 4 minimal invasive therapy.

Typing mistakes:

Page 4 and 5 Table 1

qSOFA, “point” must be corrected

Author Response

We would like to thank you for your constructive criticism, nice comments, and time spent analyzing the manuscript that helped us to improve our paper

The article requires minor revisions and some numbers and calculations must be checked.

Page 2 line 57 and 58 in fact 62.2% Escherichia coli was not among the positive cultures, but among the urine cultures. It must be corrected.

REPLY. Thank you very much for pointing out this. We changed this in abstract and full text

Pages 2 lines 58 to 61 the results must be corrected according to Huang-Tseng scale results of the table 3 and of the page 6 line 170.

REPLY. Thank you very much for pointing out this. We do apologize for this mistake. We changed this result in abstract according to table 3. Modifications were done to the analysis on Table 3 and were changed accordingly in full text.

Page 4 line 66: 62.2% Escherichia coli was not among the positive cultures, but among the urine cultures. This must be corrected.

REPLY. Thank you very much for pointing out this. We changed this in abstract and full text.

Page 4 and 5 Table 1: the number of Type1 Huang Scale must be corrected: 166 (29.1%) instead of 168, as the authors said in their answer (166 instead of 168 for type 1).

REPLY. Thank you very much for pointing out this. We do apologize for this mistake. The correct number of type 1 patients was indeed 168 (29.5%) and was modified in all tables. The mistake was in type 4 Huang, were 2 patients were incorrectly classified as type 4. The total number of type 4 Huang was 50, instead of 52 and this modification was corrected on text and in every table. Additionally, univariate and multivariate analysis (table 3) was modified due to changes in type 4 patients and corrected analysis was included in this revision.

Page 5 Table 2:

the total number of E. coli is 346 and not 356 (62.2% of 556 urine cultures; with 356 E. coli the total number of urine cultures is 566).

REPLY. Thank you very much for pointing out this. We do apologize for this mistake. We changed that

The percentage of Klebsiella spp in Arabia Saudia is 35.7 instead of 25.7

REPLY. Thank you very much for pointing out this. We do apologize for this mistake. We changed that

The percentage of ESBL agents in Mexico is 65.9% instead of 66.6%

REPLY. Thank you very much for pointing out this. We do apologize for this mistake. We changed that

Page 5 lines 159-160 The percentage of E.coli and K.pneumoniae are not reported to the positive cultures but to the total of urine cultures (negative included). It must be corrected.

REPLY. Thank you very much for pointing out this. We do apologize for this mistake. We changed that

Page 6 Table 3: Fever > 38.3°C instead of “<”

REPLY. Thank you very much for pointing out this. We do apologize for this mistake. We changed that

The number of Type1 Huang Scale in the ESBL group must be corrected: 60 (20.6%) instead of 62

REPLY. Thank you very much for pointing out this. We do apologize for this mistake. As clarified before, Huang type 1 are 168 patients and not 166. Total ESBL among Type 1 was in fact 62 (21.3%) and was modified accordingly in every table and text. The mistake regarding Huang Classification was in type 4 and that was the reason why n=572 and not the correct sample size of n=570.

Page 7 Lines 187-192 and table 5

I don’t understand how were the percentages calculated. For example, line 188, in Type 1 EPN there are 61 patients in the ESBL group, 9 of them received a conservative treatment. 9/61 = 14.8% instead of 5.4%. All the percentage of table 5 must be changed.

REPLY. Thank you very much for pointing out this. We do apologize for this mistake. We recalculated the percentages and re-checked analysis. Modifications were done accordingly and percentages were calculated considering ESBL and non-ESBL on each therapeutic modality. Example: In conservative management (n=66), Type 1 Huang reported 9/20 ESBL (45%), Type 2 6/20 (30%), Type 1 non-ESBL were 36/56 (78.3%).

Page 7 table 5

The total number of patients with ESBL is 290 instead of 291 in the table 3 and the number of patients without ESBL is 280 instead of 279 in the table 3.

REPLY. Thank you very much for pointing out this. We re-checked data and re-do analysis accordingly. Table 5 was modified and corrections were made as requested.

The total number of patients with Huang Scale: type2 is 145 instead of 144 in table 1, type 3a 107 instead of 109 in table 1, and type 3b 100 instead of 99 in table 1.

REPLY. Thank you very much for pointing out this. We apologize for this problem in Table 5. As mentioned before, data was re-checked and re-analyzed. Important changes were reported after modifications were made. Table 1-5 were analyzed and we confirm data is consistent among tables and text as well.

Some P values must be checked: 0.44 instead of 0.88 for type 3B conservative, 0.56 instead of 0.9 for type 4 conservative, 0.74 instead of 0.64 for type 4 minimal invasive therapy.

REPLY. Thank you very much for pointing out this. We do apologize for this mistake. We appreciate your valuable suggestions. As mentioned before, all table 5 was re-analyzed. Data is now consistent among all tables and text.

Typing mistakes:

Page 4 and 5 Table 1

qSOFA, “point” must be corrected

REPLY. Thank you very much for pointing out this. We do apologize for this mistake. We changed that